# wMAN: Weakly-supervised Moment Alignment Network for Text-based Video Segment Retrieval

## Abstract

Given a video and a sentence, the goal of weakly-supervised video moment retrieval is to locate the video segment which is described by the sentence without having access to temporal annotations during training. Instead, a model must learn how to identify the correct segment (*i.e.* moment) when only being provided with video-sentence pairs. Thus, an inherent challenge is automatically inferring the latent correspondence between visual and language representations. To facilitate this alignment, we propose our Weakly-supervised Moment Alignment Network (wMAN) which exploits a multi-level co-attention mechanism to learn richer multimodal representations. The aforementioned mechanism is comprised of a Frame-By-Word interaction module as well as a novel Word-Conditioned Visual Graph (WCVG). Our approach also incorporates a novel application of positional encodings, commonly used in Transformers, to learn visual-semantic representations that contain contextual information of their relative positions in the temporal sequence through iterative message-passing. Comprehensive experiments on the DiDeMo and Charades-STA datasets demonstrate the effectiveness of our learned representations: our combined wMAN model not only outperforms the state-of-the-art weakly-supervised method by a significant margin but also obtains an improvement of 10% for the Recall@1 accuracy metric over strongly-supervised state-of-the-art methods on the DiDeMo dataset.

## 1 Introduction

Video understanding has been a mainstay of artificial intelligence research. Recent work has sought to better reason about videos by learning more effective spatio-temporal representations (Tran et al., 2015; Carreira & Zisserman, 2017). The *video moment retrieval* task, also known as *text-to-clip retrieval*, combines language and video understanding to find activities described by a natural language sentence. The main objective of the task is to identify the video segment within a longer video that is most relevant to a sentence. This requires a model to learn the mapping of correspondences (alignment) between the visual and natural language modalities.

In the strongly-supervised setting, existing methods (Hendricks et al., 2017; Chen et al., 2018; Ghosh et al., 2019) generally learn joint visual-semantic representations by projecting video and language representations into a common embedding space and leverage provided temporal annotations to learn regressive functions (Gao et al., 2017) for localization. However, such temporal annotations are often ambiguous and expensive to collect. Mithun et al. (2019) seeks to circumvent these problems by proposing to address this task in the weakly-supervised setting where only full video-sentence pairs are provided as weak labels. However, the lack of temporal annotations renders the aforementioned approaches infeasible. In their approach (Figure 1a), Mithun et al. (2019) proposes a Text-Guided Attention (TGA) mechanism to attend on segment-level features w.r.t. the sentence-level representations. However, such an approach treats the segment-level visual representations as independent inputs and ignores the contextual information derived from other segments in the video. More importantly, it does not exploit the fine-grained semantics of each word in the sentence. Consequently, existing methods are not able to reason about the latent alignment between the visual and language representations comprehensively.

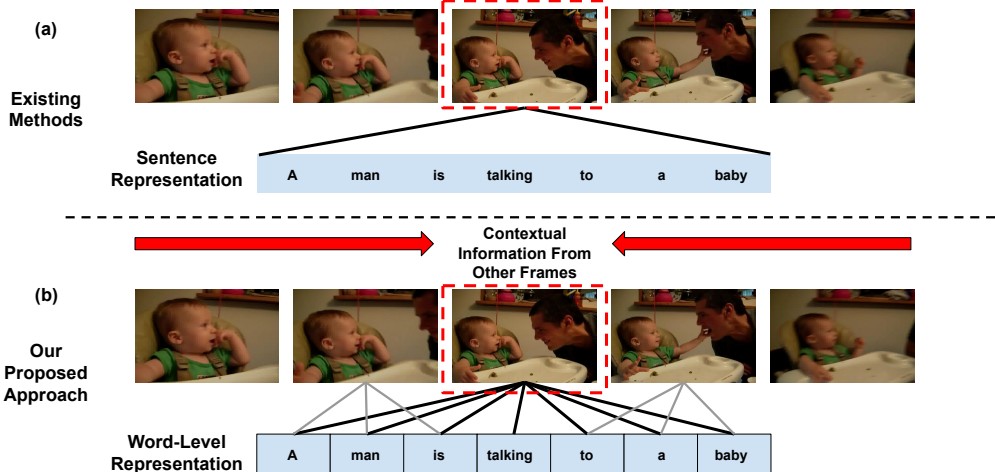

Figure 1: Given a video and a sentence, our aim is to retrieve the most relevant segment (the red bounding box in this example). Existing methods consider video frames as independent inputs and ignore the contextual information derived from other frames in the video. They compute a similarity score between the segment and the entire sentence to determine their relevance to each other. In contrast, our proposed approach aggregates contextual information from all the frames using graph propagation and leverages fine-grained frame-by-word interactions for more accurate retrieval. (Only some interactions are shown to prevent overcrowding the figure.)

In this paper, we take another step towards addressing the limitations of current weakly-supervised video moment retrieval methods by exploiting the fine-grained temporal and visual relevance of each video frame to each word (Figure 1b). Our approach is built on two core insights: 1) The temporal occurrence of frames or segments in a video provides vital visual information required to reason about the presence of an event; 2) The semantics of the query are integral to reasoning about the relationships between entities in the video. With this in mind, we propose our Weakly-Supervised Moment Alignment Network (wMAN). An illustrative overview of our model is shown in Figure 2. The key component of wMAN is a multi-level co-attention mechanism that is encapsulated by a Frame-by-Word (FBW) interaction module as well as a Word-Conditioned Visual Graph (WCVG). To begin, we exploit the similarity scores of all possible pairs of visual frame and word features to create frame-specific sentence representations and word-specific video representations. The intuition is that frames relevant to a word should have a higher measure of similarity as compared to the rest. The word representations are updated by their word-specific video representations to create visual-semantic representations. Then a graph (WCVG) is built upon the frame and visual-semantic representations as nodes and introduces another level of attention between them. During the message-passing process, the frame nodes are iteratively updated with relational information from the visual-semantic nodes to create the final temporally-aware multimodal representations. The contribution of each visual-semantic node to a frame node is dynamically weighted based on their similarity. To learn such representations, wMAN also incorporates positional encodings (Vaswani et al., 2017) into the visual representations to integrate contextual information about their relative positions. Such contextual information encourages the learning of temporally-aware multimodal representations.

To learn these representations, we use a Multiple Instance Learning (MIL) framework that is similar in nature to the Stacked Cross Attention Network (SCAN) model (Lee et al., 2018). The SCAN model leverages image region-by-word interactions to learn better representations for image-text matching. In addition, the WCVG module draws inspiration from the Language-Conditioned Graph Network (LCGN) by Hu et al. (2019) which seeks to create context-aware object features in an image. However, the LCGN model works with sentence-level representations, which does not account for the semantics of each word to each visual node comprehensively. wMAN also distinguishes itself from the above-mentioned models by extracting temporally-aware multimodal representations from videos and their corresponding descriptions, whereas SCAN and LCGN only work on images.

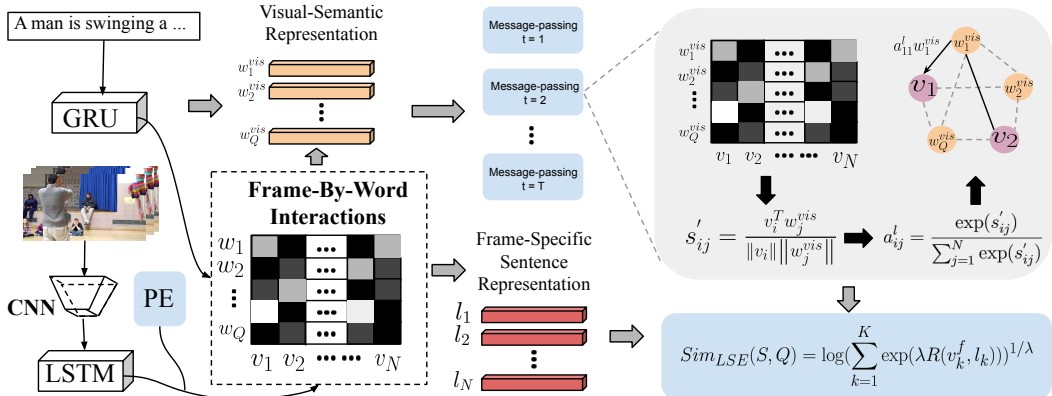

Figure 2: An overview of our combined wMAN model which is trained end-to-end. We use the outputs of the GRU as word representations where its inputs are word embeddings. The visual representations are the outputs of the LSTM unit where its inputs are the extracted features from a pretrained CNN. The visual representations are concatenated with positional encodings to integrate contextual information about their relative positions in the sequence. Our model consists of a two-stage multimodal interaction mechanism - Frame-By-Word Interactions and the WCVG.

The contributions of our paper are summarized below:

- We propose a simple yet intuitive MIL approach for weakly-supervised video moment retrieval from language queries by exploiting fine-grained frame-by-word alignment.

- Our novel Word-Conditioned Visual Graph learns richer visual-semantic context through a multi-level co-attention mechanism.

- We introduce a novel application of positional embeddings in video representations to learn temporally-aware multimodal representations.

To demonstrate the effectiveness of our learned temporally-aware multimodal representations, we perform extensive experiments over two datasets, Didemo (Hendricks et al., 2017) and Charades-STA (Gao et al., 2017), where we outperform the state-of-the-art weakly supervised model by a significant margin and strongly-supervised state-of-the-art models on some metrics.

## 2 RELATED WORK

Most of the recent works in video moment retrieval based on natural language queries (Hendricks et al., 2017; Ghosh et al., 2019; Xu et al., 2019; Zhang et al., 2019; Chen et al., 2018; Yuan et al., 2019; Chen & Jiang, 2019; Chen et al., 2019; Ge et al., 2019) are in the strongly-supervised setting, where the provided temporal annotations can be used to improve the alignment between the visual and language modalities. Among them, the Moment Alignment Network (MAN) introduced by Zhang et al. (2019) utilizes a structured graph network to model temporal relationships between candidate moments, but one of the distinguishing factors with our wMAN is that our iterative message-passing process is conditioned on the multimodal interactions between frame and word representations. The TGN (Chen et al., 2018) model bears some resemblance to ours by leveraging frame-by-word interactions to improve performance. However, it only uses a single level of attention which is not able to infer the correspondence between the visual and language modalities comprehensively. In addition, we reiterate that all these methods train their models using strong supervision, whereas we address the weakly supervised setting of this task.

There are also a number of closely-related tasks to video moment retrieval such as temporal activity detection in videos. A general pipeline of proposal and classification is adopted by various temporal activity detection models (Xu et al., 2017; Zhao et al., 2017; Shou et al., 2016) with the temporal proposals learnt by temporal coordinate regression. However, these approaches assume you are provided with a predefined list of activities, rather than an open-ended list provided via natural

language queries at test time. Methods for visual phrase grounding also tend to be provided with natural language queries as input (Chen et al., 2017; Liu et al., 2017; Faghri et al., 2018; Nam et al., 2017; Karpathy & Fei-Fei, 2015; Plummer et al., 2018), but the task is performed over image regions to locate a related bounding box rather than video segments to locate the correct moment.

# 3 WEAKLY-SUPERVISED MOMENT ALIGNMENT NETWORK

In the video moment retrieval task, given a ground truth video-sentence pair, the goal is to retrieve the most relevant video moment related to the description. The weakly-supervised version of this task we address can be formulated under the multiple instance learning (MIL) paradigm. When training using MIL, one receives a bag of items, where the bag is labeled as a positive if at least one item in the bag is a positive, and is labeled as a negative otherwise. In weakly-supervised moment retrieval, we are provided with a video-sentence pair (*i.e.*, a bag) and the video segments are the items that we must learn to correctly label as relevant to the sentence (*i.e.*, positive) or not. Following Mithun et al. (2019), we assume sentences are only associated with their ground truth video, and any other videos are negative examples. To build a good video-sentence representation, we introduce our Weakly-Supervised Moment Alignment Network (wMAN), which learns context-aware visual-semantic representations from fine-grained frame-by-word interactions. As seen in Figure 2, our network has two major components - (1) representation learning constructed from the Frame-By-Word attention and Positional Embeddings (Vaswani et al., 2017), described in Section 3.1, and (2) a Word-Conditioned Visual Graph where we update video segment representations based on context from the rest of the video, described in Section 3.2. These learned video segment representations are used to determine their relevance to their corresponding attended sentence representations using a LogSumExp (LSE) pooling similarity metric, described in Section 3.3.

## 3.1 LEARNING TIGHTLY COUPLED MULTIMODAL REPRESENTATIONS

In this section we discuss our initial video and sentence representations which are updated with contextual information in Section 3.2. Each word in an input sentence is encoded using GloVe embeddings (Pennington et al., 2014) and then fed into a Gated Recurrent Unit (GRU) (Cho et al., 2014). The output of this GRU is denoted as $W = \{w_1, w_2, \cdots, w_Q\}$ where $Q$ is the number of words in the sentence. Each frame in the input video is encoded using a pretrained Convolutional Neural Network (CNN). In the case of a 3D CNN this actually corresponds to a small chunk of sequential frames, but we shall refer to this as a frame representation throughout this paper for simplicity. To capture long-range dependencies, we feed the frame features into a Long Short-Term Memory (LSTM) (Hochreiter & Schmidhuber, 1997). The latent hidden state output from the LSTM are concatenated with positional encodings (described below) to form the initial video representations, denoted as $V = \{v_1, v_2, ..., v_N\}$ where $N$ is the number of frame features for video $V$.

**Positional Encodings (PE).** To provide some notion of the relative position of each frame we include the PE features which have been used in language tasks like learning language representations using BERT (Devlin et al., 2018; Vaswani et al., 2017). These PE features can be thought of similar to the temporal endpoint features (TEF) used in prior work for strongly supervised moment retrieval task (*e.g.*, Hendricks et al. (2017)), but the PE features provide information about the temporal position of each frame rather than the rough position at the segment level. For the desired PE features of dimension $d$, let $pos$ indicates the temporal position of each frame, $i$ is the index of the feature being encoded, and $M$ is a scalar constant, then the PE features are defined as:

$$PE_{pos,i} = \begin{cases} \sin(pos/M^{i/d}) & \text{if } i \text{ is even} \\ \cos(pos/M^{i/d}) & \text{otherwise.} \end{cases} \tag{1}$$

Through experiments, we found the hyper-parameter $M = 10,000$ works well for all videos. These PE features are concatenated with the LSTM encoded frame features at corresponding frame position before going to the cross-modal interaction layers.

### 3.1.1 FRAME-BY-WORD INTERACTION

Rather than relating a sentence-level representation with each frame as done in prior work (Mithun et al., 2019), we aggregate similarity scores between all frame and word combinations from the input

video and sentence. These Frame-By-Word (FBW) similarity scores are used to compute attention weights to identify which frame and word combinations are important for retrieving the correct video segment. More formally, for $N$ video frames and $Q$ words in the input, we compute:

$$s_{ij} = \frac{v_i^T w_j}{\|v_i\| \|w_j\|} \text{ where } i \in [1, N] \text{ and } j \in [1, Q].$$ (2)

Note that $v$ now represents the concatenation of the video frame features and the PE features.

**Frame-Specific Sentence Representations.** We obtain the normalized relevance of each word w.r.t. to each frame from the FBW similarity matrix, and use it to compute attention for each word:

$$a_{ij} = \frac{\exp(s_{ij})}{\sum_{j=1}^{Q} \exp(s_{ij})}.$$ (3)

Using the above-mentioned attention weights, a weighted combination of all the words are created, with correlated words to the frame gaining high attention. Intuitively, a word-frame pair should have a high similarity score if the frame contains a reference to the word. Then the frame-specific sentence representation emphasizes words relevant to the frame and is defined as:

$$l_i = \sum_{j=1}^{Q} a_{ij} w_j.$$ (4)

Note that these frame-specific sentence representations don't participate in the iterative message-passing process (Section 3.2). Instead, they are used to infer the final similarity score between a video segment and the query (Section 3.3).

**Word-Specific Video Representations.** To determine the normalized relevance of each frame w.r.t. to each word, we compute the attention weights of each frame:

$$a_{ij}' = \frac{\exp(s_{ij})}{\sum_{i=1}^{N} \exp(s_{ij})}.$$ (5)

Similarly, we attend to the visual frame features with respect to each word by creating a weighted combination of visual frame features determined by the relevance of each frame to the word. The formulation of each word-specific video-representation is defined as:

$$f_j = \sum_{i=1}^{N} a_{ij}' v_i.$$ (6)

These word-specific video representations are used in our Word-Conditioned Visual Graph, which we will discuss in the next section.

## 3.2 Word-Conditioned Visual Graph Network

Given the sets of visual representations, word representations and their corresponding word-specific video representations, the WCVG aims to learn temporally-aware multimodal representations by integrating visual-semantic and contextual information into the visual features. To begin, the word representations are updated with their corresponding video representations to create a new visual-semantic representation $w_j^{vis}$ by concatenating each word $w_j$ and the word-specific video representation $f_j$. Intuitively, the visual-semantic representations not only contain the semantic context of each word but also a summary of the video with respect to each word. A fully connected graph is then constructed with the visual features $v_i$ and the embedded attention of visual-semantic representations $w_j^{vis}$ as nodes.

**Iterative Word-Conditioned Message-Passing** The iterative message-passing process introduces a second round of FBW interaction similar to that in Section 3.1.1 to infer the latent temporal correspondence between each frame $v_i$ and visual-semantic representation $w_j^{vis}$. The goal is to update the representation of each frame $v_i$ with the video context information from each word-specific video representation $w_j^{vis}$. To realize this, we first learn a projection $W_1$ followed by a ReLU of $w_j^{vis}$ to

obtain a new word representation to compute a new similarity matrix $s'_{ij}$ on every message-passing iteration, namely, we obtain a replacement for $w_j$ in Eq. (2) via $w'_j = ReLU(W_1(w_j^{vis}))$.

**Updates of Visual Representations** During the update process, each visual-semantic node sends its message (represented by its representation) to each visual node weighted by their edge weights. The representations of the visual nodes at the t-th iteration are updated by summing up the incoming messages as follows:

$$v_i^t = W_2(concat\{v_i^{t-1}; \sum_{j=1}^{Q} a_{ij}^l w'_j\}), \tag{7}$$

where $a_{ij}$ is obtained by applying Eq. (3) to the newly computed FBW similarity matrix $s'_{ij}$, and $W_2$ is a learned projection to make $v_i^t$ the same dimensions as the frame-specific sentence representation $l_i$ (refer to Eq. (4) ) which are finally used to compute a sentence-segment similarity score.

## 3.3 MULTIMODAL SIMILARITY INFERENCE

The final updated visual representations $V^T = \{v_1^T, v_2^T, \cdots, v_V^T\})$ are used to compute the relevance of each frame to its attended sentence-representations. A segment is defined as any arbitrary continuous sequence of visual features. We denote a segment as $S = \{v_1^T, \cdots, v_K^T\}$ where $K$ is the number of frame features contained within the segment $S$. We adopt the LogSumExp (LSE) pooling similarity metric used in SCAN (Lee et al., 2018), to determine the relevance each proposal segment has to the query:

$$Sim_{LSE}(S, Q) = \log(\sum_{k=1}^{K} \exp(\lambda R(v_k^f, l_k)))^{1/\lambda} \text{ where } R(v_k, l_k) = \frac{v_k^T l_k}{\|v_k\| \|l_k\|}. \tag{8}$$

$\lambda$ is a hyperparameter that weighs the relevance of the most salient parts of the video segment to the corresponding frame-specific sentence representations. Finally, following Mithun et al. (2019), given a triplet $(X^+, Y^+, Y^-)$, where $(X^+, Y^+)$ is a positive pair and $(X^+, Y^-)$ a negative pair, we use a margin-based ranking loss $L_T$ to train our model which ensures the positive pair's similarity score is better than the negative pair's by at least a margin. Our model's loss is then defined as:

$$L_{total} = \sum_{(V^+, Q^+)} \{\sum_{Q^-} L_T(V^+, Q^+, Q^-) + \sum_{V^-} L_T(Q^+, V^+, V^-)\}. \tag{9}$$

$Sim_{LSE}$ is used as the similarity metric between all pairs. At test time, $Sim_{LSE}$ is also used to rank the candidate temporal segments generated by sliding windows, and the top scoring segments will the localized segments corresponding to the input query sentence.

## 4 EXPERIMENTS

We evaluate the capability of wMAN to accurately localize video moments based on natural language queries without temporal annotations on two datasets - DiDeMo and Charades-STA. On the DiDeMo dataset, we adopt the mean Intersection-Over-Union (IOU) and Recall@N at IOU threshold = $\theta$. Recall@N represents the percentage of the test sliding window samples which have a overlap of at least $\theta$ with the ground-truth segments. mIOU is the average IOU with the ground-truth segments for the highest ranking segment to each query input. On the Charades-STA dataset, only the Recall@N metric is used for evaluation.

### 4.1 DATASETS

**Charades-STA** The Charades-STA dataset is built upon the original Charades [Sigurdsson et al. (2016)] dataset which contains video-level paragraph descriptions and temporal annotations for activities. Charades-STA is created by breaking down the paragraphs to generate sentence-level annotations and aligning the sentences with corresponding video segments. In total, it contains 12,408 and 3,720 query-moment pairs in the training and test sets respectively. For fair comparison with the weakly model TGA (Mithun et al., 2019), we use the same non-overlapping sliding windows of sizes 128 and 256 frames to generate candidate temporal segments.

Table 1: Moment retrieval performance comparison on the Charades-STA test set. (a) contains representative results of strongly-supervised methods reported in prior works while (b) reports the performance of weakly-supervised methods including our approach.

| Method | Training Supervision | iou = 0.3 | | | iou = 0.5 | | | iou = 0.7 | | |
|---|---|---|---|---|---|---|---|---|---|---|
| | | R@1 | R@5 | R@10 | R@1 | R@5 | R@10 | R@1 | R@5 | R@10 |
| (a) CTRL (Gao et al., 2017) | Strong | - | - | - | 23.63 | 58.92 | - | 8.89 | 29.52 | - |
| MLVI (Xu et al., 2019) | Strong | 54.7 | 95.6 | 99.2 | 35.6 | 79.4 | 93.9 | 15.8 | 45.4 | 62.2 |
| MAN (Zhang et al., 2019) | Strong | - | - | - | 46.53 | 86.23 | - | 22.72 | 53.72 | - |
| (b) TGA (Mithun et al., 2019) | Weak | 29.68 | 83.87 | 98.41 | 17.04 | 58.17 | 83.44 | 6.93 | 26.80 | 44.06 |
| wMAN (ours) | Weak | **48.04** | **89.01** | **99.57** | **31.74** | **72.17** | **86.58** | **13.71** | **37.58** | **45.16** |
| Upper Bound | - | - | - | - | 99.84 | - | - | 88.17 | - | - | 46.80 |

**DiDeMo** The videos in the Distinct Describable Moments (DiDeMo) dataset are collected from Flickr. The training, validation and test sets contain 8395, 1065 and 1004 videos respectively. Each query contains the temporal annotations from at least 4 different annotators. Each video is limited to a maximum duration of 30 seconds and equally divided into six segments with five seconds each. With the five-second segment as basic temporal unit, there are 21 possible candidate temporal segments for each video. These 21 segments will used to compute the similarities with the input query and the top scored segment will be returned as the localization result.

## 4.2 IMPLEMENTATION DETAILS

For fair comparison, we utilize the same input features as the state-of-the-art method (Mithun et al., 2019). Specifically, the word representations are initialized with GloVe embeddings and fine-tuned during the training process. For the experiments on DiDeMo, we use the provided mean-pooled visual frame and optical flow features. The visual frame features are extracted from the fc7 layer of VGG-16 [Simonyan & Zisserman (2014)] pretrained on ImageNet [Deng et al. (2009)]. The input visual features for our experiments on Charades-STA are C3D [Tran et al. (2015)] features. We adopt an initial learning rate of $1e-5$ and a margin= $0.5$ used in our model's triplet loss (Eq. 9). In addition, we use three iterations for the message-passing process. Our model is trained end-to-end using the ADAM optimizer.

## 4.3 RESULTS

### 4.3.1 CHARADES-STA

The results in Table 1 show that our full model outperforms the TGA model by a significant margin on all metrics. In particular, the Recall@1 accuracy when IOU = 0.7 obtained by our model is almost doubled that of TGA. It is notable that we observe a consistent trend of the Recall@1 accuracies improving the most across all IOU values. This not only demonstrates the importance of richer joint visual-semantic representations for accurate localization but also the superior capability of our model to learn them. Our model also performs comparably to the strongly-supervised MAN model on several metrics.

To better understand the contributions of each component of our model, we present a comprehensive set of ablation experiments in Table 2. Note that our combined wMAN model is comprised of the FBW and WCVG components as well as the incorporation of PEs. The results obtained by our FBW variant demonstrate that capturing fine-grained frame-by-word interactions is essential to inferring the latent temporal alignment between these two modalities. More importantly, the results in the second row (FBW-WCVG) show that the second stage of multimodal attention, introduced by the WCVG module, encourages the augmented learning of intermodal relationships. Finally, we also observe that incorporating positional encodings into the visual representations (FBW-WCVG + PE) are especially helpful in improving Recall@1 accuracies for all IOU values. We provide results for a model variant that include TEFs which encode the location of each video segment. In Table 2, our experiments show that TEFs actually hurt performance slightly. Our model variant with PEs (FBW-

Table 2: Charades-STA ablation experiment results on a held-out validation set. In the table, * indicates the same number of model parameters as the combined wMAN model.

| Method | iou = 0.3 | | | iou = 0.5 | | | iou = 0.7 | | |
|---|---|---|---|---|---|---|---|---|---|
| | R@1 | R@5 | R@10 | R@1 | R@5 | R@10 | R@1 | R@5 | R@10 |
| FBW | 41.41 | **93.79** | **99.23** | 26.91 | **72.19** | 85.97 | 10.83 | 34.85 | 45.20 |
| FBW-WCVG | 43.99 | 90.85 | 99.19 | 28.24 | 70.70 | 86.14 | 11.64 | 34.85 | 45.20 |
| FBW-WCVG + TEF | 43.99 | 88.03 | 98.99 | 28.01 | 69.19 | 86.01 | 11.20 | 35.29 | 44.45 |
| FBW-WCVG * | 42.18 | 91.09 | 99.19 | 27.47 | 70.58 | 86.09 | 11.52 | 34.97 | 45.11 |
| FBW-WCVG + PE (wMAN) | **46.05** | 91.25 | 99.19 | **29.00** | 69.46 | **86.26** | **13.30** | **36.99** | **45.32** |

Table 3: Moment retrieval performance comparison on the DiDeMo test set. (a) contains representative results of strongly-supervised methods reported in prior works while (b) reports the performance of weakly-supervised methods including our approach.

| | Method | Training Supervision | R@1 | R@5 | mIOU |
|---|---|---|---|---|---|
| **(a)** | MCN Hendricks et al. (2017) | Strong | 28.10 | 78.21 | 41.08 |
| | TGN (Chen et al., 2018) | Strong | 28.23 | 79.26 | 42.97 |
| **(b)** | TGA (Mithun et al., 2019) | Weak | 12.19 | 39.74 | 24.92 |
| | wMAN | Weak | **38.07** | **63.94** | **38.37** |
| | Upper Bound | - | 74.75 | 100.00 | 96.05 |

WCVG + PE) outperforms the model variant with TEFs (FBW-WCVG + TEF) on all of the metrics. We theorize that the positional encodings aid in integrating temporal context and relative positions into the learned visual-semantic representations. This makes it particularly useful for Charades-STA since its videos are generally much longer.

To gain insights into the fine-grained interactions between frames and words, we provide visualizations in Figure 3. Our model is able to determine the most salient frames with respect to each word relatively well. In both examples, we observe that the top three salient frames with respect to each word are generally distributed over the same subset of frames. This seems to be indicative of the fact that our model leverages contextual information from all video frames as well as words in determining the salience of each frame to a specific word.

### 4.3.2 DiDeMo

Table 3 reports the results on the DiDeMo dataset. In addition to reporting the state-of-the-art weakly supervised results, we also include the results obtained by strongly-supervised methods. It can be observed that our model outperforms the TGA model by a significant margin, even tripling the Recall@1 accuracy achieved by them. This demonstrates the effect of learning richer joint visual-semantic representations on the accurate localization of video moments. In fact, our full model outperforms the strongly-supervised TGN and MCN models on the Recall@1 metric by approximately 10%.

We observe a consistent trend in the ablation studies (Table 4) as with those of Charades-STA. In particular, through comparing the ablation models FBW and FBW-WCVG, we demonstrate the effectiveness of our multi-level co-attention mechanism in WCVG where it improves the Recall@1 accuracy by a significant margin. Similar to our observations in Table 2, PEs help to encourage accurate latent alignment between the visual and language modalities, while TEFs fail in this aspect.

## 5 CONCLUSION

In this work, we propose our weakly-supervised Moment Alignment Network with Word-Conditioned Visual Graph which exploits a multi-level co-attention mechanism to infer the latent alignment between visual and language representations at fine-grained word and frame level. Learning context-aware visual-semantic representations helps our model to reason about the temporal occurrence of an event as well as the relationships of entities described in the natural language query.

Table 4: DiDeMo ablation experiment results on the validation set. In the table, * indicates the same number of model parameters as the combined wMAN model.

| Method | R@1 | R@5 | MIOU |
|---|---|---|---|
| FBW | 30.19 | **66.74** | 39.06 |
| FBW-WCVG | 39.93 | 66.53 | 39.19 |
| FBW-WCVG + TEF | 37.55 | 66.36 | 39.11 |
| FBW-WCVG * | 39.06 | 66.31 | 39.05 |
| FBW-WCVG + PE (wMAN) | **41.62** | 66.57 | **39.20** |

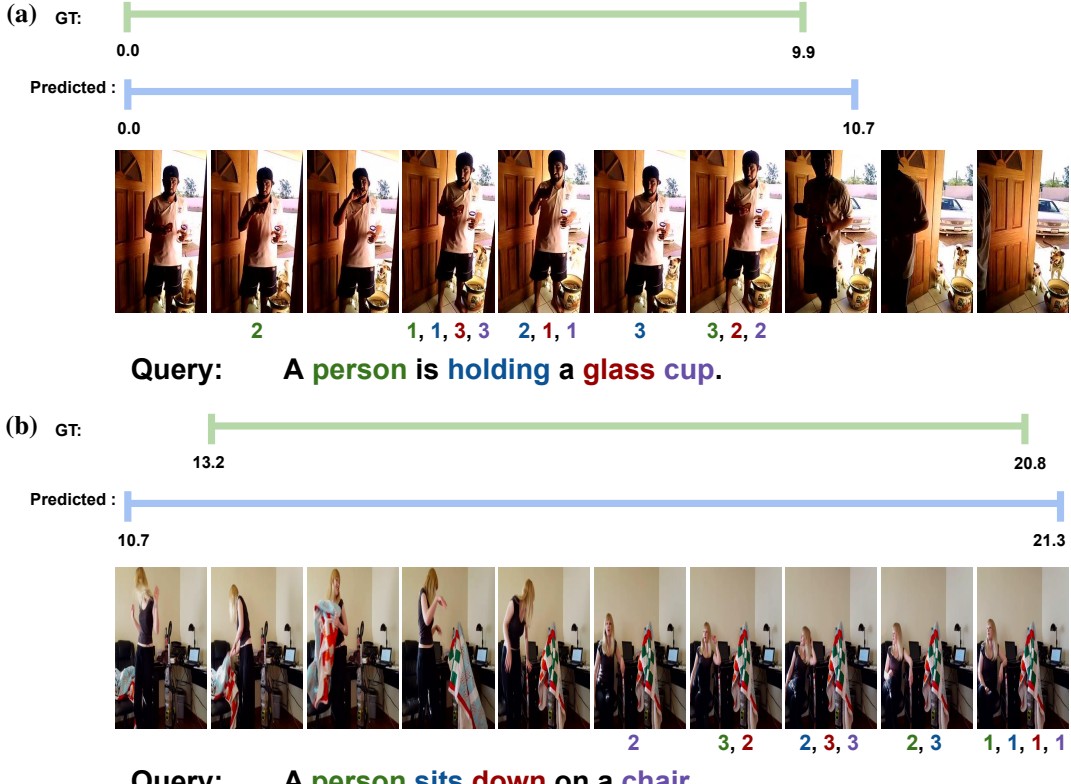

Figure 3: Visualization of the final relevance weights of each word in the query with respect to each frame. Here, we display the top three weights assigned to the frames for each phrase. The colors of the three numbers (1,2,3) indicate the correspondence to the words in the query sentence. We also show the ground truth (GT) temporal annotation as well as our predicted weakly localized temporal segments in seconds. The highly correlated frames to each query word generally fall into the GT temporal segment in both examples.

Finally, our experimental results empirically demonstrate the effectiveness of such representations on the accurate localization of video moments.

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

## A  EXPERIMENTS FOR NUMBER OF MODEL PARAMETERS ON CHARADES-STA

Table 5: Moment retrieval performance comparison on the Charades-STA test set. In this table, we display the number of model parameters as well as the results achieved on the Charades-Sta Test Set.

| Method | Number of Model Parameters | iou = 0.3 | | | iou = 0.5 | | | iou = 0.7 | | |
|---|---|---|---|---|---|---|---|---|---|---|
| | | R@1 | R@5 | R@10 | R@1 | R@5 | R@10 | R@1 | R@5 | R@10 |
| TGA (Mithun et al., 2019) | 3M | 29.68 | 83.87 | 98.41 | 17.04 | 58.17 | 83.44 | 6.93 | 26.80 | 44.06 |
| TGA (Mithun et al., 2019) | 19M | 27.36 | 77.58 | 99.03 | 14.38 | 59.97 | 85.83 | 5.24 | 30.40 | 44.67 |
| LCGN  (Hu et al., 2019) | 152M | 35.81 | 82.93 | 99.09 | 19.25 | 65.11 | 85.19 | 7.12 | 32.90 | 43.63 |
| FBW | 3M | 38.13 | 90.49 | 99.48 | 24.73 | 69.42 | 85.29 | 9.91 | 34.32 | 43.94 |
| FBW | 20M | 38.33 | **91.10** | 99.43 | 24.51 | 69.18 | 86.31 | 10.24 | 33.17 | 44.40 |
| FBW + WCVG | 18M | 42.84 | 88.05 | 99.54 | 27.60 | 70.00 | **86.61** | 11.47 | 34.30 | 44.73 |
| wMAN (ours) | 18M | **48.04** | 89.01 | **99.57** | **31.74** | **72.17** | 86.58 | **13.71** | **37.58** | **45.16** |

In Table  5, we show the comparisons of the different methods with different number of model parameters.  While wMAN has 18M parameters as compared to 3M parameters in TGA, the performance gains are not simply attributed to the number of model parameters.  We increase the dimensions of visual and semantic representations as well as corresponding fully-connected layers in the TGA model which leads to a total of 19M parameters. Despite having more parameters than wMAN, it still does significantly worse on all metrics. We also provide results obtained by a direct adaptation of the Language-Conditioned Graph Network (LCGN), which is designed to work on the image level for VQA as well. While LCGN leverages attention over the words in the natural language query, the computed attention is only conditioned on the entire sentence without contextual information derived from the objects' visual representations. In contrast, the co-attention mechanism in our combined wMAN model is conditioned on both semantic and contextual visual information derived from words and video frames respectively. LCGN is also a lot more complicated and requires significantly more computing resources than wMAN. Despite possessing much more parameters than wMAN, it is still not able to achieve comparable results to ours.

## B  ABLATION ON NUMBER OF MESSAGE PASSING ROUNDS

Table 6: Charades-STA ablation experiment results on the held-put validation set.

| Number of Message Passing Rounds | iou = 0.3 | | | iou = 0.5 | | | iou = 0.7 | | |
|---|---|---|---|---|---|---|---|---|---|
| | R@1 | R@5 | R@10 | R@1 | R@5 | R@10 | R@1 | R@5 | R@10 |
| 2 | 43.39 | 86.14 | 99.13 | 15.18 | 68.89 | 86.14 | 13.10 | 36.14 | 45.18 |
| 3 | **46.05** | **91.25** | **99.19** | **29.00** | 69.46 | 86.26 | **13.30** | **36.99** | **45.32** |
| 4 | 43.71 | 88.07 | 99.15 | 15.31 | 68.94 | **86.53** | 13.10 | 36.50 | 45.24 |

Table 7: DiDeMo ablation experiment results on the validation set.

| Number of Message Passing Rounds | R@1 | R@5 | MIOU |
|---|---|---|---|
| 2 | 40.04 | **66.57** | 39.14 |
| 3 | **41.62** | **66.57** | **39.20** |
| 4 | 40.11 | 66.36 | 39.06 |

In this section, we include ablation results on the number of message-passing rounds required to learn effective visual-semantic representations. In our experiments, we have found that three rounds work best on both Charades-Sta and DiDeMo.

# C ABLATION ON TEST SETS

Table 8: Charades-STA ablation experiment results on the test set.

| Method | iou = 0.3 | | | iou = 0.5 | | | iou = 0.7 | | |
|---|---|---|---|---|---|---|---|---|---|
| | R@1 | R@5 | R@10 | R@1 | R@5 | R@10 | R@1 | R@5 | R@10 |
| FBW | 41.47 | **95.99** | 99.56 | 26.74 | 72.68 | 86.50 | 10.37 | 34.97 | 44.89 |
| FBW + PE | 45.77 | 95.59 | 99.41 | 29.68 | **74.92** | 86.23 | 12.71 | 37.47 | **45.16** |
| FBW-WCVG | 42.95 | 90.16 | 99.35 | 27.79 | 70.53 | **86.66** | 11.26 | 34.83 | 45.00 |
| FBW-WCVG + PE (wMAN) | **48.04** | 89.01 | **99.57** | **31.74** | 72.17 | 86.58 | **13.71** | **37.58** | **45.16** |

Table 9: DiDeMo ablation experiment results on the test set.

| Method | R@1 | R@5 | MIOU |
|---|---|---|---|
| FBW | 26.43 | 63.88 | 38.49 |
| FBW + PE | 20.67 | 63.44 | **38.58** |
| FBW-WCVG | 36.47 | 63.83 | 38.36 |
| FBW-WCVG + PE (wMAN) | **38.07** | **63.94** | 38.37 |

The results in Tables 8 and 9 emphasize the importance of both PEs and WCVG in our approach. We observe the best performances when PEs are integrated into the visual-semantic representations learned by the WCVG. The consistency of these performance gains across both datasets seems indicative of the increased capability of our model to learn temporally-aware visual-semantic representations.

