# OpenReview forum: "wMAN: WEAKLY-SUPERVISED MOMENT ALIGNMENT NETWORK FOR TEXT-BASED VIDEO SEGMENT RETRIEVAL"
_ICLR.cc/2020/Conference — Reject_

### Official Review · AnonReviewer1 · 2019-10-25
**Official Blind Review #1**

**Rating:** 6

**Review:**


Summary:
This paper proposes a method for aligning an input text with the frames in a video that correspond to what the text describes in a weakly supervised way. The authors propose a combination of a “Frame-By-Word” (FBW) representation and a Word-Conditioned Visual Graph (WCVG). The proposed method outperforms the weakly supervised baseline presented in the paper in experiments by a large margin. In addition, it quantitatively performs close to previous strongly supervised methods.


Pros:
+ New Word-Conditioned Visual Graph representation
+ Outperforms weakly supervised baseline
+ Ablation study of the moving parts
+ Interesting use of positional embeddings for multi-modal learning

Weaknesses / comments:
- What is the processing speed of the method compared to the baseline?
The proposed method makes multiple comparisons while computing the attention weights over all words and frames. Does this cause the method to be slower than the baseline? If so, how much slower is it?
Answers to these questions can help readers to keep in mind the trade-off of the proposed method for achieving the accuracy presented in the paper.


- Number of parameters comparison with baseline:
Did the authors make sure to have similar number of model parameters for the baselines and the proposed method? Maybe I missed it, but I couldn’t see a mention of this anywhere. It would be useful to state this so that readers are sure that it’s not the number of parameters that is helping the method.


- Assumption that sentences are only associated with its ground truth video:
The authors mention that they have the same assumption as Mithun et al., 2019. Can this assumption be detrimental if the dataset does not follow it? Say there are sentences in the dataset that could describe segments in multiple videos. Could this assumption lead to suboptimal representation learning / relationship learning for words / video frames?

- Determining the size of the sliding window:
From reading the paper, it looks like the sliding window used for computing the word / frame relationships has to be manually defined. This seems a bit suboptimal for the generalizability of this method. Do the authors have any comments on this?

- Can this model be supervised? If so, how does it compare to the supervised baselines?
The authors point out that their weakly supervised method performs close to the strongly supervised previously proposed. This is a nice finding, however, have the authors try to answer the question of what would happen if the proposed model is supervised? Will the proposed model outperform the strongly supervised baselines? Or at least perform the same?

Conclusion:
In conclusion, the proposed method makes sense and it has been shown to empirically outperforms a previous weakly supervised baseline. The authors also provide an ablation study of the moving parts to show that the entire pipeline is important to achieve the highest performance in the hardest setting. It would be nice if the authors successfully answer / address the questions / concerns mentioned above in the rebuttal.

**Experience Assessment:**

I have read many papers in this area.

**Review Assessment: Checking Correctness Of Derivations And Theory:**

N/A

**Review Assessment: Checking Correctness Of Experiments:**

I carefully checked the experiments.

**Review Assessment: Thoroughness In Paper Reading:**

I read the paper thoroughly.

---

> ### Author Response · Authors · 2019-11-10
> **Response to Reviewer 1**
>
> Thank you for your review. We address your concerns below.
>
> - What is the processing speed of the method compared to the baseline?
>
> In our measurements, the TGA method takes approximately 4.725s to process a single query. In contrast, the FBW module and our combined model (wMAN) take 4.329s and 8.102s respectively. As evident, the difference in processing time is not significant. However, if speed were really a huge concern, using the FBW module alone would provide a much faster processing time as well as improved results over the TGA method. For reference, the TGA model obtains an average recall accuracy of 49.8% while the FBW module achieves 57.0%.
>
> - Number of parameters comparison with baseline
>
> The TGA model contains about 3M parameters while wMAN contains 18M parameters. However, the large performance gains are not directly attributed to the increase in parameters. To prove this, we increase the dimensions of feature representations as well as relevant fully-connected layers in TGA such that the total number of parameters becomes 19M. We evaluate this model on Charades-Sta and the results are provided in Table 5 in the appendix. As evident, even with more parameters than our model, it still does substantially worse than ours. To add on to this, our direct adaptation of the Language-Conditioned Graph Network (also provided in Table 5), which has 152M parameters, also yields results inferior to ours. Finally, we also decrease the number of parameters in the FBW module alone to 3M and its performance gain over the TGA model is still significant.
>
> - Assumption that sentences are only associated with its ground truth video
>
> This assumption does add random noise to the training process. However, there is a higher probability of assigning a non-relevant sentence that does not correspond to its ground truth video than to the contrary. With that said, this assumption is also often used in tasks such as image-sentence retrieval or phrase grounding.
>
> - Determining the size of the sliding window
>
> We definitely agree with this. In response to this, we simply adopted the same candidate proposals adopted by the baseline and prior work for fair comparisons. However, this is an interesting avenue for future work and we are actually exploring possible ways of replacing these manually-defined sliding window mechanism with an efficient subwindow search algorithm.
>
> - Can this model be supervised? If so, how does it compare to the supervised baselines?
>
> Our model should be easily generalizable to the strongly-supervised setting. Due to time constraints and other commitments, we do not have the time to adapt it to the strongly-supervised setting for this rebuttal, but this is an interesting future work direction.
>
> We hope that we have addressed your concerns satisfactorily. Please let us know if you have any further concerns or questions.

---

### Official Review · AnonReviewer3 · 2019-10-31
**Official Blind Review #3**

**Rating:** 6

**Review:**

The paper proposed a weakly-supervised wMAN model for moment localization in untrimmed videos. Only the video-level annotation is available for training, and the goal is retrieving the video segment described by the sentence. The proposed model explored to utilize better context information and captured the relation between video and sentence/word via graph neural networks. In particular, instead of modeling the context information between the sentence and each video frame, wMAN tried to learn the representation with multi-level and co-attention, which considers all possible pairs between the word and the frame. The proposed model was evaluated on two publicly-available dataset and achieved reasonable results.

Pros:
- Weakly-supervised method for video moment localization is a reasonable and important direction.
- wMAN explicitly utilized multi-level context information between the sentence and the video frame, and used the graph neural network and the message passing to model the representation. I think this is a reasonable direction.
- wMAN is evaluated with two publicly available datasets, and is compared with state-of-the-art methods and other "oracle" baselines. The performance is impressive and could be a better baseline for the future work.

Cons:
- wMAN model the relation for all possible pairs of the word and the video frame. However, if the video is quite long, say 10 minutes, 30 minutes, or even few hours, will the method still be efficient and effective?
- When building the relation between the word and the frame, is there any emphasis on verb, some particular word, or self-learned attention? For some particular word, say "people" and "cup", won't it have strong connection with many frames? But for some of the words, say "hold" and "sits", could it play a more important role?
- Followed by previous question, in the qualitative results, it seems the boundary parts of the predicted video segments are less accurate. Is it because some of the words case these false positive results? What do you think the reason is?
- Experimental results: I suggest the author to provide more ablation analysis to the experiment section. For example, the full model of wMAN works better than FBW on R@1, but worse on R@5 and R@10. Is there a particular reason about this? PE seems to be important for wMAN, and the authors provides few sentences analysis about this, but I don't think I fully understand this part. Another problem is that there is only few qualitative results, and in both these two examples, predicted results cover the GT segments. Is this always the case for wMAN? Why? Some failure cases could also be very helpful.
- Less technical comments: The paper writing is fine to me, but I don't like the typesetting. I suggest to put the model figure more close to the methodology section and the qualitative results on page 8.

Overall, I think the paper is marginal above the accept line.

**Experience Assessment:**

I have read many papers in this area.

**Review Assessment: Checking Correctness Of Derivations And Theory:**

I carefully checked the derivations and theory.

**Review Assessment: Checking Correctness Of Experiments:**

I carefully checked the experiments.

**Review Assessment: Thoroughness In Paper Reading:**

I read the paper at least twice and used my best judgement in assessing the paper.

---

> ### Author Response · Authors · 2019-11-10
> **Response to Reviewer 3**
>
> Thank you for your review. We address your concerns below.
>
> - wMAN model the relation for all possible pairs of the word and the video frame. However, if the video is quite long, say 10 minutes, 30 minutes, or even few hours, will the method still be efficient and effective?
>
> Computing effective video representations for long videos efficiently is still an unsolved problem in computer vision. There is a lot of ongoing work in this area. With this said, based on the observed memory requirements of our proposed approach during inference, the efficiency and effectiveness of our method should be scalable to videos lasting a few minutes. As mentioned before, reasoning about videos lasting a few hours efficiently and effectively is still an unsolved research topic. However, with increased computational resources, there is no reason to believe that our method is not scalable to such videos. One possible solution is to reduce the sampling rate of video frame. Another option is to break the video into smaller parts and localize within each part individually. Finding a way to reason about long videos and natural language effectively from a low frame sampling rate in this task provides an interesting avenue for future work.
>
> - When building the relation between the word and the frame, is there any emphasis on verb, some particular word, or self-learned attention?
>
> The motivation behind the frame-by-word interaction mechanism in our approach is that it encourages the model to learn the association between words and action sequences in videos. Words such as ‘hold’ and ‘sits’ definitely play a much more important role in localizing the relevant temporal segment in videos. For example, in Figure 3b, we observe that the top 3 weights assigned to each frame for ‘person’ and ‘chair’ generally occur in tandem with ‘sits’ and ‘down’. This demonstrates that our model learns the association between verbs and entities via self-learned attention. This is consistent with our observations in Figure 3a as well.
>
> - Followed by previous question, in the qualitative results, it seems the boundary parts of the predicted video segments are less accurate.
>
> One possible reason is that we are using non-overlapping segments as proposals on Charades-Sta to facilitate fair comparison with prior work. Given that these proposals have static boundaries, it will cause the boundary parts of the candidate proposals to be less accurate.
>
> - Experimental results: I suggest the author to provide more ablation analysis to the experiment section.
>
> It appears that contextual cues generally help to improve retrieval accuracy on harder settings such as higher IOU thresholds and Recall@1 accuracy. Using just the FBW module leads to better performance only on the lowest IOU threshold and Recall@5 and Recall@10 accuracies. We observe the same consistency in our ablation experiments on DiDeMo as well. We hypothesize that these cues help to make our model more discriminative in harder settings which is arguably more practical for real-world applications such as in video search engines. Finally, the overall performance of wMAN is better than that of the FBW module. If we average the scores, we obtain 57.0% and 58.2% for the FBW module and wMAN respectively.
>
> - Less technical comments
>
> We will update the next version of the paper with the necessary clarifications and modifications.
>
> We hope that we have addressed your concerns satisfactorily. Please let us know if you have any further concerns or questions.

---

### Official Review · AnonReviewer2 · 2019-11-03
**Official Blind Review #2**

**Rating:** 3

**Review:**

This work presents a model for text based video clip (video moments or text-to-clip) retrieval. The goal is to identify a video segment within a longer video that is most relevant to an input sentence. Authors propose a new model based on a weakly-supervised training approach. This model does not require explicit temporal annotations to align text and video, but it only needs as an input the full video and sentence pairs.

Key aspects of the model are: i) A coattention step frame-by-word and word-by-frame that produces the basic embeddings of the model, which is enriched with positional information, and ii) A contextual step that aggregates contextual information from all the frames using graph propagation. Afterwards, they use a  LogSumExp pooling strategy to score similarity among the input sentence and video frame.

The main contribution of the paper is incremental (specially respect to Mithun et al., 2019), I do not see a ground-breaking contribution. One of the main novelties with respect to previous text-to-clip models is the use of co-attention schemes at the level of words and frames. However, the idea of co-attention at different grain-levels have been proposed before. Actually, while the model makes an extensive use of frame-to-word encoding, it is not clear to me what is the role of the word-to-video representation in Eqs. 5 and 6.

In general, the paper is well written. The experimental evaluation is convincing. However, it is not clear why authors change the structure of the evaluation among the experiments. As an example, for the experiments in Charades-STA dataset, they include scores for different IOUs levels, but they do not repeat this for DiDeMo dataset. Similarly, for DiDeMo dataset, results in Table 3 are for the test set, while the ablation study in Table 4 is for the validation set. I will recommend to standardize the evaluations.

Another comment is that in several experiment best performance is obtained using just the FBW module, it will be interesting to further analyze why the contextual cues hurt performance in some cases, maybe at least a qualitative analysis. Also, in some part of the papers, authors state that the proposed model does better than strongly-supervised state-of-the-art methods on some metrics, looking all the reported tables, I do not think that this is the case. Authors show qualitative results about cases where the model perform well, it will be good to also analyze failure cases, actually, according to the final scores, there is still lot of cases that the model can't handle properly.

I rate the paper as borderline, but there is not such a rating at ICLR 2020, so I will lean to weak reject.

**Experience Assessment:**

I have read many papers in this area.

**Review Assessment: Checking Correctness Of Derivations And Theory:**

I assessed the sensibility of the derivations and theory.

**Review Assessment: Checking Correctness Of Experiments:**

I carefully checked the experiments.

**Review Assessment: Thoroughness In Paper Reading:**

I read the paper at least twice and used my best judgement in assessing the paper.

---

> ### Author Response · Authors · 2019-11-10
> **Response to Reviewer 2**
>
> Thank you for your review. We address your concerns below.
>
> - The main contribution of the paper is incremental (specially respect to Mithun et al., 2019), I do not see a ground-breaking contribution. One of the main novelties with respect to previous text-to-clip models is the use of co-attention schemes at the level of words and frames.
>
> We address this in the general response.
>
> - Actually, while the model makes an extensive use of frame-to-word encoding, it is not clear to me what is the role of the word-to-video representation in Eqs. 5 and 6.
>
> We describe its purpose in the beginning paragraph of Section 3.2.  It is concatenated with the word embedding to create a new visual-semantic representation, which is next used to update the visual representations iteratively during message-passing as shown in equation 7. The intuition is that the word-specific representations help to convey contextual visual information derived from other video frames. We will update the next version of the paper to make it clearer.
>
> - However, it is not clear why authors change the structure of the evaluation among the experiments.
>
> We adopt the same evaluate metrics and practices as prior work to enable direct comparison. Scores for different IOU thresholds are used on the Charades-Sta dataset while scores for only IOU=0.5 are used on DiDeMo.  With respect to the ablation experiments not being evaluated on the test set, we followed the standard protocol of finetuning hyperparameters and evaluating model components on the validation set. This is more realistic in the real world where we usually do not have access to the test set in practice. We do have the ablation results on the test set too but we left them out due to space constraints. However, we have added them to the Appendix in Tables 8 and 9.
>
> - It will be interesting to further analyze why the contextual cues hurt performance in some cases, maybe at least a qualitative analysis.
>
> It appears that contextual cues generally help to improve retrieval accuracy on harder settings such as higher IOU thresholds and Recall@1 accuracies. Using just the FBW module leads to better performance only on the lowest IOU threshold and Recall@5 and Recall@10 accuracies. We observe the same consistency in our ablation experiments on DiDeMo as well. We hypothesize that these cues help to make our model more discriminative in harder settings which is arguably more practical for real-world applications such as in video search engines. Finally, the overall performance of wMAN is better than that of the FBW module. If we average the scores, we obtain 57.0% and 58.2% for the FBW module and wMAN respectively.
>
> - In some part of the papers, authors state that the proposed model does better than strongly-supervised state-of-the-art methods on some metrics
>
> In Table 3, we show that we outperform the strongly-supervised methods by 10% on the Recall@1 metric.  We have clarified this in the paper.
>
> We hope that we have addressed your concerns satisfactorily. Please let us know if you have any further concerns or questions.

---

### Official Review · AnonReviewer4 · 2019-11-04
**Official Blind Review #4**

**Rating:** 3

**Review:**

Overview:
The authors proposed a weakly-supervised method to localize video moments given text queries.  The model builds multi-level relational graphs among pairs of word and video frame, and the graph is used to aggregate visual-semantic feature for each word and each frame. Then the attentive features are used to localize the sentence query in videos by calculating the similarity of words and frames. In summary, the proposed weakly-supervised Moment Alignment Network (wMAN) utilizes a multi-level co-attention mechanism to learn richer multimodal representations for language based video retrieval..

Pros:
1. Significant performance improvement on Didemo and Charades-STA datasets. The authors achieved very good performance on both dataset, even higher than some of the full-supervision methods, such as CTRL and MLVI.

Cons:
1. The overall novelty of the proposed methods is limited. Essentially, the key points of the model is hierarchical visual semantic co-attention.,which is proposed originally in [Hierarchical Question-Image Co-Attention
for Visual Question Answering], although the original application is VQA in image domain. So in this way, the novelty is only marginal.
2. Paper writing can be improved. Figure 2 shows the overall structure of the model, however, the caption doesn't explain all the notations in the figure, such as WCVG, and the equations. Additionally, the reference is very far away from Figure 2, which makes the whole paper hard to read.
3. For evaluation part, one important ablation study is missing: the number of steps T for message passing. This eval is important, as it shows the necessity of using "multi-level" attention.

Minor comments:
1. Make the caption of Figure 2 self-explainable, e.g. the meaning of LSE.
2. There is a "word-conditioned" visual graph network, why not the other way, "frame-conditioned" semantic graph net and iterate over it?


**Experience Assessment:**

I have published one or two papers in this area.

**Review Assessment: Checking Correctness Of Derivations And Theory:**

I assessed the sensibility of the derivations and theory.

**Review Assessment: Checking Correctness Of Experiments:**

I assessed the sensibility of the experiments.

**Review Assessment: Thoroughness In Paper Reading:**

I read the paper thoroughly.

---

> ### Author Response · Authors · 2019-11-10
> **Response to Reviewer 4**
>
> Thank you for your review. We address your concerns below.
>
> 1) This is addressed in the general response.
>
> 2) We will update the next version of the paper with the necessary clarifications to the caption and modifications.
>
> 3) We have updated the submission with an ablation study of how the number of message-passing steps affects the performance of our proposed approach. They are included in Section B of the appendix. In our experiments across both Charades-Sta and DiDeMo, we have observed that 3 steps work the best.
>
> We hope that we have addressed your concerns satisfactorily. Please let us know if you have any further concerns or questions.

---

### Author Response · Authors · 2019-11-10
**General Response**

We sincerely thank all reviewers for their time and effort in reviewing our submission! Your reviews have provided us with interesting insights for future work. Reviewers agree that weakly-supervised text-based video moment localization is an important research direction (R3) and that our proposed approach leverages a new Word-Conditioned Visual Graph to aggregate contextual information from both words and video frames (R1 and R3). Our model also incorporates positional embeddings for multimodal learning (R1) which has not been used before in non-Transformers based approaches to the best of our knowledge. Besides outperforming the weakly-supervised baseline by a significant margin (R2 and R4), it also performs very comparably to strongly-supervised state-of-the-art methods, even outperforming them on some metrics on the DiDeMo dataset.

The submission has been updated to include more ablation experiment results, especially in the appendix section. To begin, we would like to review the contributions of our paper and benefits of our approach over prior work before addressing the specific questions individually in the sections below.

We agree that the general idea of co-attention has been proposed before but in a different context as mentioned. It has not been successfully applied to weakly-supervised text-based video retrieval, which has its own unique set of challenges compared to image VQA. Our model, though also based on co-attention, is very different from predecessors and experimentally works much better.

Our hierarchical co-attention mechanism has some key differences from that of the Video Question Answering (VQA) model Hierarchical Question-Image Co-Attention for Visual Question Answering (HieCoAttVQA). One key difference is that we incorporate Positional Encodings (PEs), typically used in Transformers for language modeling, in our multimodal interaction mechanism. As shown in our ablation experiments, these PEs helps to enrich the capability of our model in modeling long-range dependencies between video segments as opposed to simply increasing the dimensions of the visual representations (shown in Tables 2 and 4). To the best of our knowledge, this is a novel use of PEs for multimodal learning in non-Transformers based approaches.

Our co-attention mechanism also introduces a novel word-conditioned visual graph. During the message-passing process, our graph-based approach iteratively updates our visual representations with not only semantic information but with contextual information from other video frames as well, derived from word and word-specific video representations respectively. In contrast, the co-attention mechanism in the VQA paper simply alternates attending to the image and question representations separately.

We would like to reiterate that while co-attention has been proposed in other contexts (e.g. images), the exact method of accomplishing this is crucial for the task that we are addressing, especially in modeling long range dependencies in the much harder domain of videos. The most telling indication of this is the performance difference between our and the TGA (Weakly-Supervised Baseline)model, which also uses co-attention,  that we are comparing to. Our model achieves 3x and 2x accuracies of the TGA  model on DiDeMo and Charades-Sta on the hardest setting respectively. This demonstrates the importance of the different components in our approach.

To further emphasize the importance of the exact implementation of the co-attention mechanism, we provide results obtained from an adaptation of the Language-Conditioned Graph Network (LCGN) that is also designed to work on the image domain for VQA. Similarly, it employs a co-attention mechanism to reason about relationships between objects and words. The results are included in Table 5 in the appendix. On Charades-Sta, the LCGN model generally performs better than the TGA model. However, the obtained results are still vastly inferior to those achieved by wMAN. Finally, our approach serves as a good baseline of comparison for future work in this direction.

---

### Decision · Program_Chairs · 2019-12-19

**Decision:**

Reject

**Comment:**

This paper proposes a method for aligning an input text with the frames in a video that correspond to what the text describes in a weakly supervised way. The main technical contribution of the paper is the use of co-attention at different abstraction levels.

Among the four reviewers, one reviewer advocates for the paper while the others find this paper to be a borderline reject paper. Reviewer3 who was initially positive about the paper, during the discussion period, expressed that he/she wants to downgrade his/her rating to weak reject after reading the other reviewers' comments and concerns. The main concern of the reviewers is that the contribution of the paper incremental, particularly since the idea of co-attention has been used in many different area in other context. The authors responded to this in the rebuttal that the proposed approach incorporate different components such as Positional Encodings and is different from prior work, and that they experimentally perform superior compared to other co-attention usages such as LCGN. Although the AC understands the authors response, the majority of the reviewers are still not fully convinced about the contribution and their opinion stay opposed to the paper.